# Molecular Characterization of Medulloblastoma in a Patient with Neurofibromatosis Type 1: Case Report and Literature Review

**DOI:** 10.3390/diagnostics11040647

**Published:** 2021-04-02

**Authors:** Marco Ranalli, Alessandra Boni, Anna Maria Caroleo, Giada Del Baldo, Martina Rinelli, Emanuele Agolini, Sabrina Rossi, Evelina Miele, Giovanna Stefania Colafati, Luigi Boccuto, Iside Alessi, Maria Antonietta De Ioris, Antonella Cacchione, Rossella Capolino, Andrea Carai, Sabina Vennarini, Angela Mastronuzzi

**Affiliations:** 1Department of Pediatrics, Sapienza University, Viale Regina Elena 324, 00161 Rome, Italy; marco.ranalli@uniroma1.it (M.R.); lale.boni88@gmail.com (A.B.); evelina.miele@opbg.net (E.M.); 2Department of Onco-Hematology and Cell and Gene Therapy, Bambino Gesù Children’s Hospital (IRCCS), 00165 Rome, Italy; annamaria.caroleo@opbg.net (A.M.C.); giada.delbaldo@opbg.net (G.D.B.); iside.alessi@opbg.net (I.A.); mantonietta.deioris@opbg.net (M.A.D.I.); antonella.cacchione@opbg.net (A.C.); 3Laboratory of Medical Genetics, Bambino Gesù Children’s Hospital (IRCCS), 00165 Rome, Italy; martina.rinelli@opbg.net (M.R.); emanuele.agolini@opbg.net (E.A.); 4Pathology Unit, Department of Laboratories, Bambino Gesù Children’s Hospital (IRCCS), 00165 Rome, Italy; sabrina.rossi@opbg.net; 5Neuroradiology Unit, Department of Imaging, Bambino Gesù Children’s Hospital (IRCCS), 00165 Rome, Italy; gstefania.colafati@opbg.net; 6School of Nursing, College of Behavioral, Social and Health Sciences Healthcare Genetics Interdisciplinary Doctoral Program, Clemson University, Clemson, SC 29631, USA; lboccut@clemson.edu; 7Medical Genetics Unit, Bambino Gesù Children Hospital, Bambino Gesù Children’s Hospital (IRCCS), 00165 Rome, Italy; rossella.capolino@opbg.net; 8Neurosurgery Unit, Department of Neurosciences, Bambino Gesù Children’s Hospital (IRCCS), 00165 Rome, Italy; andrea.carai@opbg.net; 9Proton Therapy Center, Hospital of Trento, Azienda Provinciale per I Servizi Sanitari (APSS), 38122 Trento, Italy; sabina.vennarini@apss.tn.it

**Keywords:** medulloblastoma, neurofibromatosis type 1, pediatric cancer, brain tumor

## Abstract

Brain tumors are the most common solid neoplasms of childhood. They are frequently reported in children with Neurofibromatosis type 1 (NF1). The most frequent central nervous system malignancies described in NF1 are optic pathway gliomas and brainstem gliomas. Medulloblastoma (MB) in NF1 patients is extremely rare, and to our knowledge, only 10 cases without molecular characterization are described in the literature to date. We report the case of a 14-year-old girl with NF1 that came to our attention for an incidental finding of a lesion arising from cerebellar vermis. The mass was completely resected, revealing a localized classic medulloblastoma (MB), subgroup 4. She was treated as a standard-risk MB with a dose-adapted personalized protocol. The treatment proved to be effective, with minor toxicity. Brain and spine MRI one year after diagnosis confirmed the complete remission of the disease. To our knowledge, this is the only case of MB reported in a patient with NF1 with molecular characterization by the methylation profile. The association between NF1 and MB, although uncommon, may not be an accidental occurrence.

## 1. Introduction

Neurofibromatosis type 1 (NF1, OMIM #162200) is a relatively common genetic syndrome, with a prevalence ranging between 1/3000 and 1/6000 worldwide [1,2,3,4]. The disorder affects multiple systems with major cutaneous, neurologic, and orthopedic manifestations, which lead to significant morbidity and mortality [5,6,7,8]. NF1 patients have an increased risk of developing malignancies and a life expectancy of about 10–15 years shorter than the general population [9]. NF1 diagnosis is primarily based on the National Institutes of Health diagnostic criteria (Table 1): 97% of patients with NF1 meet such criteria by eight years of age and all by 20 years of age [10]. These criteria usually appear in the following predictable sequence of symptoms: café-au-lait macules, axillary freckling, Lisch nodules, and neurofibromas. The characteristic osseous lesions typically appear within the first year of life, and the mean age ranges from three to six years at diagnosis of optic gliomas [10].

A vast number of different pathogenic variants in the *NF1* gene have been described [11,12,13,14,15]. Large deletions involving the *NF1* locus (chromosomal region 17q11.2) have been associated with a more severe phenotype, including developing neurofibromas at an earlier age, having a lower mean intelligence quotient, abnormal facial features, and an elevated risk for malignant peripheral nerve sheath tumors [16,17,18].

Malignancies are the leading cause of death in NF1 [19,20]. Compared to the general population, individuals with NF1 are two-to-three times more likely to develop cancer of the esophagus, stomach, colon, or lungs and three–seven times more likely to develop cancer of the liver, thyroid, ovary, breast, malignant melanoma, non-Hodgkin’s lymphoma, or chronic myeloid leukemia in their lifetime. They are also 15 times more likely to develop small intestine tumors and 20 times more likely to develop bone cancer [21], specifically before 50 years of age [22].

Children under 10 years old are the most susceptible to the development of intracranial tumors that occur in 20% of individuals with NF1 [23]. The most common central nervous system (CNS) tumor associated with NF1 is the optic pathway glioma (OPG), reported in 15–20% of children affected with this condition [24,25]. The second-most frequent brain tumor in individuals with NF1 is brainstem glioma (BSG), which is an indolent neoplasm with a good prognosis [26,27,28] that arises in slightly older children (mean age is seven years) and are often incidentally discovered on neuroimaging studies [28,29,30]. Similar to OPGs, these tumors are usually low-grade gliomas and, in particular pilocytic astrocytomas. Gliomas can also be found in other locations, such as the temporal lobe, cerebellum, thalamus, basal ganglia, or spinal cord [31], and a larger part are asymptomatic. NF1 patients can also develop high-grade gliomas (HGG). These are uncommon in children with NF1 but increase in prevalence during early adulthood [31], with estimates of a >50-fold increased risk relative to the general population [32]. HGG usually arise in the cerebral hemispheres.

Medulloblastoma (MB) is the most common pediatric brain tumor. Advances in molecular characterization have found significant heterogeneity among MBs, which has led to the identification of four distinct molecular subgroups included in the 2016 WHO Classification of Tumors of the Central Nervous System (wingless (WNT), sonic hedgehog (SHH), group 3, and group 4) [33,34]. Moreover, new scientific data has shown that germline pathogenic variants in cancer predisposition genes account for about 5% to 6% of all MB diagnoses [35].

The finding of MB in NF1 patients is extremely rare: to our knowledge, only ten cases without molecular characterization have been reported in the literature.

We hereby describe, to the best of our knowledge, the first molecularly characterized case of a subgroup 4 MB reported in a child with NF1.

## 2. Case Presentation

A 14-year-old mixed-race girl affected by familial NF1 (affected father) came to our attention for the incidental finding of a lesion arising from the cerebellar vermis and extending to the roof of the fourth ventricle, without detection of secondary lesions. It was revealed accidentally by a magnetic resonance imaging (MRI) that was performed for increasing headaches. The sequence of the *NF1* gene (NM_000267) on DNA isolated from the blood did not reveal germline pathogenic variants. Subsequently, her genetic profile was studied by the Multiplex Ligation-dependent Probe Amplification (MLPA) analysis for microdeletions/microduplications of the *NF1* gene (NM_000267), which detected the deletion of the entire *NF1* gene (assay performed by Multiplex Ligation-dependent Probe Amplification (MLPA) kits P081-D1-0418 and P082-C2-0419 (Mrc Holland, Amsterdam, The Netherlands) and capillary electrophoresis with Genetic Analyzer automated sequencer (Applied Biosystems, Waltham, MA, USA).

At the physical examination, the girl presented NF1 stigmatic characteristics, which included more than six diffuse pathognomonic café-au-lait spots, neurofibromas on the left wrist and shoulders, freckling in the trunk and groin region, Lisch nodules in both eyes, and Tanner puberty stage V. The neurological examination showed there were no motor or sensory deficits and no cranial nerve abnormalities.

After preoperatory evaluation that showed a localized disease, the patient underwent a complete surgical resection of the intracranial mass. A lumbar puncture (LP) was performed 15 days after surgery, and no neoplastic cells were detected.

Microscopically, the tumor showed a vaguely nodular pattern (Figure 1A), but desmoplasia was mostly absent and limited to the areas of leptomeningeal invasion (Figure 1B). Immunohistochemical analyses revealed that the tumor tissue was diffusely positive for synaptophysin and negative for *YAP1* and *GAB1*, whereas the *β-catenin* expression was limited to the cytoplasm (Figure 1C–F). Ki67 was highly expressed in the internodular areas (Figure 1G). Coherently, Sanger sequencing failed to detect pathogenic variants in the *CTNNB1* gene. The tumor was interpreted as a MB, classic type, non-WNT, and non-SHH. Gain/amplification of *MYC/MYCN*, such as a *TP53* mutation, was not found.

As previously reported, DNA methylation profiling was performed in accordance with the protocols approved by the Bambino Gesù Children’s Hospital Institutional Review Board (1556_OPBG_2018, approved on 15/01/2019) after written consent was obtained from the patient’s parents.

In the brain tumor classifier developed by Heidelberg University, Heidelberg, Germany [36], the tumor scored significantly in the “methylation class family medulloblastoma group 3 and 4, subtype VI”, reaching a raw score of 0.55 (Figure 2A) and a calibrated score of 0.98 in the “methylation class medulloblastoma, subclass group 4”. Molecular subgroups 3 and 4 that reflect non-WNT/non-SHH groups in immunohistochemistry are consistent with limited cytoplasm *β-catenin* expression and are *YAP1-* and *GAB1-*immunonegative, as previously described.

The plotted copy number variation showed the partial deletion of chromosomes 3, 8, and 16q, as well as partial chromosome 7 and 18 gains. (Figure 2B). The known deletion of the entire *NF1* gene locus was also evident (red arrow). Neither the methylation array nor the real-time PCR demonstrated a gain/amplification of *MYC/MYCN*.

Complete resected and isolated classical MB without gain/amplification of *MYC/MYCN* and *TP53* pathogenic variants allowed for the classification as a standard-risk MB. Considering her genetic syndrome, it was not possible to enroll the patient in the MB protocol. In consideration of the increased risk of secondary malignancies, she was treated with a personalized regimen according to the European indications for standard-risk MB, based on proton therapy rather than standard radiotherapy and four courses of reduced doses of vincristine, cyclophosphamide, and cisplatin. Vincristine-related neurotoxicity was reported, and two doses of vincristine were omitted.

The brain and spine MRI one year after diagnosis confirmed the complete remission of the disease.

## 3. Discussion

NF1 is an autosomal-dominant tumor predisposition syndrome caused by deletions or pathogenic variants in the *NF1* gene [9,11,12,13,14]. Neurofibromin, the product of the *NF1* gene, is a negative regulator of the *RAS/MAPK* pathway [37]. Its loss of function leads to an increased *RAS* pathway activity, with a subsequent increased proliferation and tumorigenesis, specifically in neurocutaneous tissues [38].

Brain tumors occur in about 20% of individuals with NF1 [23]. Cerebral hemisphere and posterior fossa tumors, such as astrocytomas and medulloblastomas, are uncommon in NF1 patients and occur globally at a 1% rate [39].

MBs account for about 9.2% of pediatric brain tumors [40]. The incidence of MB peaks during the first decade of life, with a higher incidence noted in children between three and four years of age and between eight and 10 years of age [40]. A risk stratification established based on the histopathological subtype, age at diagnosis, staging, residual disease, *MYC* and *TP53* status, and molecular subgrouping allows for a distinction of low-, standard-, and high-risk patients [41].

Low- and standard-risk patients are characterized by ages over three years old, the absence of metastatic and/or residual disease, histotypes other than anaplastic/large cells, the absence of *MYC* amplification, and/or *TP53* pathogenic variants. On the other side, patients less than three years of age with residual postoperative tumors, metastases, anaplastic/large cell histotypes, *MYC* amplification, and *TP53* mutations reflected high-risk MB [42]. Recently, to more accurately predict the outcome, an updated risk stratification took into account the subgroup status and genetic and cytogenetic aberrations. This new risk stratification system allocates patients to one of four risk groups: low-risk (> 90% survival), standard-risk (75–90% survival), high-risk (50–75% survival), and very high risk (< 50% survival) [42]. Four different subgroups were identified based on genetic profiling studies: Wingless (WNT), Sonic Hedgehog (SHH), Group 3, and Group 4; each of these is characterized by a specific set of demographic, clinical, and genetic features [30].

In our case, the patient was affected by the classic group 4 MB. Group 4 tumors form the largest molecular subgroup of MBs, comprising about 35% of the overall cases. Similar to SHH subgroup tumors, Group 4 tumors have a prognosis that is intermediate between WNT and Group 3 MBs. Within Group 4 MBs, significantly inferior outcomes have been observed in patients with metastatic disease or *MYCN* amplification [40].

The treatment of MB includes a combination of surgery, radiation therapy (in patients >three years old), and chemotherapy and is associated with significant morbidity, especially in the youngest patients.

MB has been reported in association with certain cancer predisposition syndromes [43], such as the MB SHH subtype with Gorlin-Goltz syndrome (*PTCH1* and *SUFU* genes) [43,44]. The elevated frequency of MBs has also been observed in patients with Turcot syndrome (*APC* gene and *MMR* genes), where colon polyps and brain tumors are associated in a context of alterations of the *WNT* pathway [45]. Similarly, numerous MBs have been reported in patients with Nijmegen syndrome as a result of defects in the DNA repair signaling pathways [46]. Few cases of MBs have been associated with Down syndrome [47]. Other rare genetic syndromes associated with MB involve pathogenic variants in the *BRCA2*, *PALB2*, *GPR161*, and *ELP* genes that were recently identified, even if cases of NF1 patients are not reported [43].

MB is very rarely associated with NF1. To the best of our knowledge, a total of ten cases are described in the scientific literature to date. In Table 2, we briefly described the principal characteristics of our case and the other 10 cases described in the literature. Information regarding the MB histology of the reported patients is very poor, and no patient, other than ours, has a known molecular subgroup.

The first case was described in 1969 by Corkill and Ross [48]. They described an eight-year-old boy with NF1 and multiple tumors (medulloblastoma, multiple neurofibromas/sarcomas, and radiation-induced thyroid carcinoma). The patient died nine years after due to the metastatic involvement of sarcomatous nerve tumors. Perilongo et al. [49] described in 1993 an eight-year-old girl with NF1 who developed four consecutively primary malignant tumors: a Wilms tumor, T-cell acute lymphoblastic leukemia, MB, and acute myeloid leukemia. Magimairajan et al. [50] described a case of NF1 with MB and immunohistochemistry for mismatch repair positivity.

In this study, we reported the eleventh case of a patient with NF1 and MB subgroup 4. This is the first case investigated by DNA methylation profiling. The association between MB and NF1 is not well-established, and further studies are needed to elucidate the possible molecular mechanisms underlying this correlation. As declared by Martinez-Lage et al. [51], this association, although uncommon, may not be an accidental occurrence, and neurofibromatosis should be included in the cancer predisposition syndrome for medulloblastoma. One of the possible second hits in individuals with NF1 is a somatic rearrangement leading to isochromosome 17q, causing the loss of the remaining wild-type allele of the *NF1* gene and triggering tumorous degeneration in the affected tissue. Such a isochromosome, 17q, has been reported in several cases of MBs belonging to the GP4 group. This possible connection may provide further validation to the mechanistic link between NF1 and MBs.

Moreover, no significant clinical-pathological correlation or molecular profile were described until now in medulloblastoma related to NF1. To the best of our knowledge, molecular methylation profile characterization was described for the first time in our case, and more cases are needed to establish if *NF1* gene pathogenic variants or deletions can drive medulloblastoma development.

In the future, it will be crucial to molecularly classify these tumors in order to better understand their biology and to allow prognostic evaluations of this peculiar patient population.

**Table 2 diagnostics-11-00647-t002:** Medulloblastoma in patients with Neurofibromatosis type 1 (NF1) described in the literature.

Article	Cases/NF1 Patients	Age at MB Diagnosis (y)	Histology	Outcome	FU	Other Findings
Corkill et al. (1969) [48]	1	8	unk	Died at 17 y because of metastatic sarcomatous nerve tumors in MB CR	9 y	Multiple neurofibromas, thyroid carcinoma, and sarcomatous nerve tumor at age of 17 y
Robles Cascallar et al. (1992) [52]	1	1	unk	unk	unk	no
Perilongo et al. (1993) [49]	1	8	unk	unk	unk	Wilms tumor, T-cell acute lymphoblastic, and myeloid leukemia
Martinez-Lage et al. (2002) [51]	1	6	Classic	Alive in CR	5 y	no
Rosenfeld et al. (2010) [32]	1/740	16	Anaplastic	Alive in CR	11 m	no
Pascual-Castrovejo et al. (2010) [39]	1/600	4	unk	Died	3 y	no
Varan et al. (2015) [53]	2/473	#1: 4 #2: 9	unk	#1: Died #2: Alive in CR	#1: 1 m #2: 11 y	#1: no #2: no
Magimairajan et al. (2016) [50]	1	unk	unk	Alive in CR	6 m	MMRD
Marinău et al. (2017) [54]	1	4	Desmoplastic	Died for metastatic progression	6 m	no
Ranalli et al. (our case)	1	14	Classic—subgroup 4	Alive in CR	6 m	no

Unk, unknown; CR, complete remission; FU, follow-up; y, years; m, months; MMRD, mismatch repair defect genes; and MB, medulloblastoma.

## Figures and Tables

**Figure 1 diagnostics-11-00647-f001:**
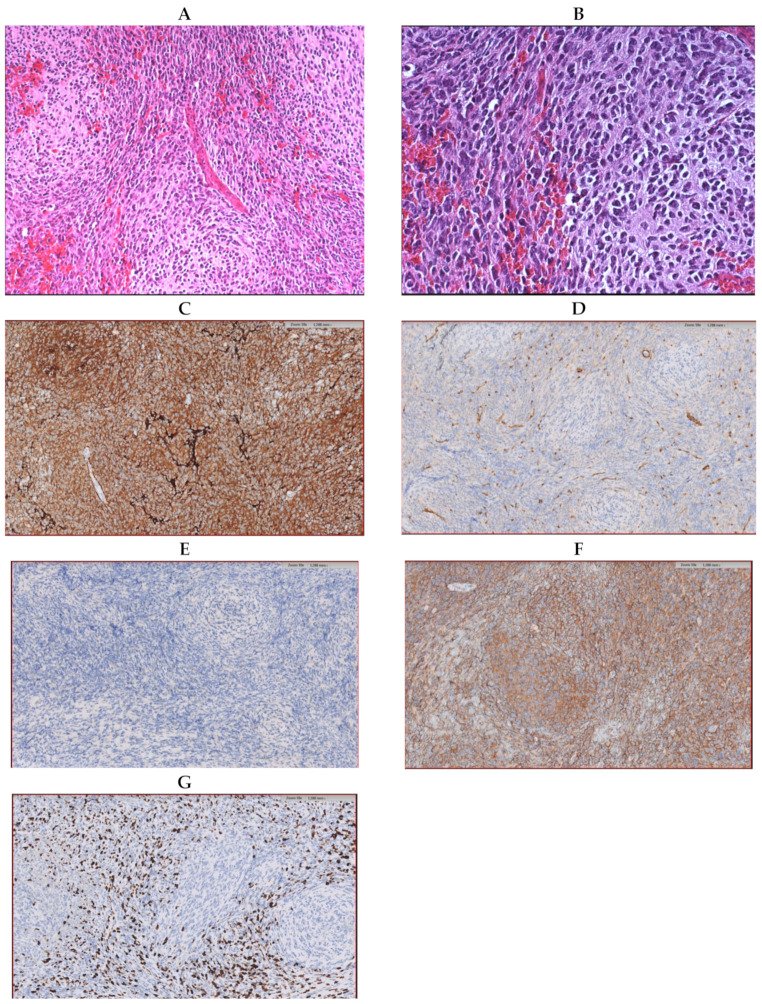
Histological samples preparation of the tumor (hematoxylin-;eosin coloration) showing a vague nodular pattern (**A**) without an increase in reticulin (**B**). Immunohistochemical samples revealing the diffused expression of synaptophysin (**C**), *YAP1* solely expressed by the endothelial cells (**D**), no expression of *GAB1* (**E**), the expression of *Beta-catenin* limited to the cytoplasm (**F**), and a high expression of *Ki67* in the internodular areas (**G**).

**Figure 2 diagnostics-11-00647-f002:**
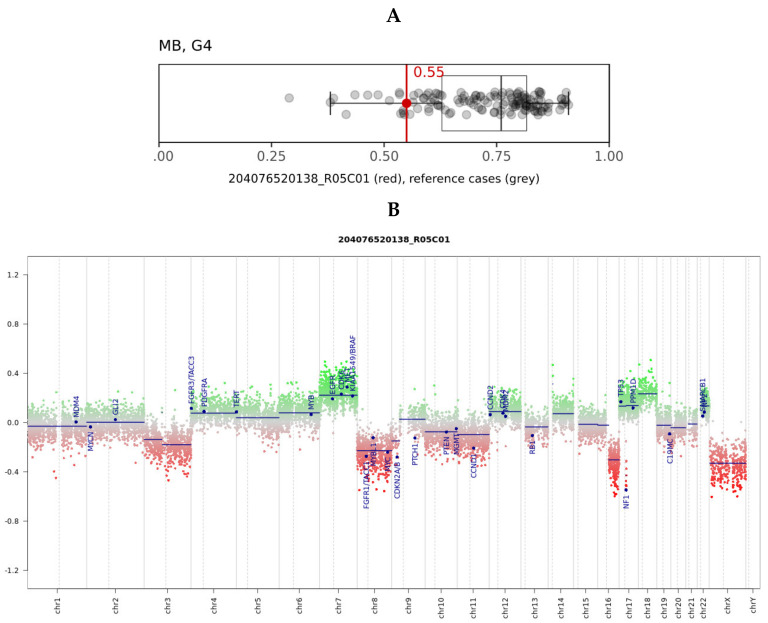
(**A**) Box and whisker plots depict the maximum raw classification scores (0.55) of the tumor sample in the methylation class “methylation class medulloblastoma, subclass group 4 (MB, G4)”, according to brain tumor classifier v11b4. Grey dots represent the reference cases in the methylation class. (**B**) Copy number variation plots were calculated from the DNA to 22 and X. Gains/amplifications represent positive (green), and losses negative (red), deviations from the baseline. Twenty-nine brain tumor-relevant genomic regions are highlighted. The red arrow points out the deleted chromosome 17q region, including the *NF1* gene.

**Table 1 diagnostics-11-00647-t001:** National Institutes of Health clinical diagnostic criteria for neurofibromatosis type 1 [6].

Presence of 2 of the Following:
1. ≥6 café-au-lait macules >5 mm in diameter in prepubertal individuals and >15 mm in postpubertal individuals
2. ≥2 neurofibromas of any type or 1 plexiform neurofibroma
3. Freckling in the axillary or inguinal regions
4. ≥2 Lisch nodules
5. Optic glioma
6. A distinctive osseous lesion, such as sphenoid wing dysplasia, or thinning of the long bone cortex, with or without pseudoarthrosis
7. First-degree relative (parent, sibling, or offspring) with NF1 based on the above criteria

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
