# Peer review of "Molecular Characterization of Medulloblastoma in a Patient with Neurofibromatosis Type 1: Case Report and Literature Review"

_diagnostics, 2021, doi:10.3390/diagnostics11040647_

Round 1

Reviewer 1 Report

Renalli et al. provide a concise and interesting case report of a GP4 medulloblastoma arising in a child with NF1. This study adds data to the growing field genetic predisposition in childhood brain tumors and potential clinical guidance for future similar cases.

I have some minor suggestions that I feel could improve the quality of the manuscript.

IHC: add relevance of YAP1/GAB1/b-catenin with respect to subgrouping

Methylation classifier results: Could you add the GP3/4 subtype classification (GP3/4 I through VIII; see Sharma et al; Acta Neuropath, 2019)– easily available through the same the MNP.org classifier used by the authors for subgroup analysis.

Patient Treatment: Could you add details of the therapy given and the changes to standard risk therapy in consideration of NF1 and the risk of secondary malignancies. This will be useful for future cases.

CNV: The apparent isochromosome 17q observed in this patient should be noted, as this is a classic feature of GP4. Perhaps this provides a connection between GP4 medulloblastoma and NF1 on 17q as a genetic predisposition.

Predisposition discussion: The authors elaborate on the study by Waszak et al (Lancet Oncol 2018), a very large study of genetic predisposition in medulloblastoma (~1000 patients), that showed the majority of genetic predispositions are in SHH, with few observed in GP4 (PALB2 and BRCA2), and that no NF1 mutations were identified.

Author Response

We are pleased that you have considered our work and provided us with very interesting comments that will enhance our paper.

Point 1: IHC: add relevance of YAP1/GAB1/b-catenin with respect to subgrouping 

Response 1: Thank you very much for your suggestion. We have underlined the connection between cytoplasmic b-catenin, YAP1 and GAB1 immunonegativity to molecular subgroup 3/4.

Point 2: Methylation classifier results: Could you add the GP3/4 subtype classification (GP3/4 I through VIII; see Sharma et al; Acta Neuropath, 2019)– easily available through the same the MNP.org classifier used by the authors for subgroup analysis.

Response 2: Thank you for your observation. Additional data about subtype GP3/4 will increase the relevance of our work. In our case, subtype VI was found, and we have added this information to the paper.

Point 3: Patient Treatment: Could you add details of the therapy given and the changes to standard risk therapy in consideration of NF1 and the risk of secondary malignancies. This will be useful for future cases

Response 3: Thank you for your recommendation and observation. We have better explained that for the genetic condition of our patient, she cannot be enrolled in a protocol. She was treated accordingly with European standard risk medulloblastoma indications based on proton-therapy (rather than standard radiotherapy) and 4 courses of reduced doses of vincristine, cyclophosphamide, and cisplatin.

Point 4: CNV: The apparent isochromosome 17q observed in this patient should be noted, as this is a classic feature of GP4. Perhaps this provides a connection between GP4 medulloblastoma and NF1 on 17q as a genetic predisposition.

Response 4: Thank you for your suggestion. We have strengthened this concept in the discussion: ”One of the possible second hits in individuals with NF1 is a somatic rearrangement leading to isochromosome 17q, causing the loss of the remaining wild-type allele of the NF1_gene and triggering the tumorous degeneration in the affected tissue. Such isochromosome 17q has been reported in several cases of MBs belonging to GP4 group. This possible connection may provide further validation to the mechanistic link between NF1 and MBs”.

Point 5: Predisposition discussion: The authors elaborate on the study by Waszak et al (Lancet Oncol 2018), a very large study of genetic predisposition in medulloblastoma (~1000 patients), that showed the majority of genetic predispositions are in SHH, with few observed in GP4 (PALB2 and BRCA2), and that no NF1 mutations were identified.

Response 5: Thank you for your comments. In accordance with your observation, in the f Waszak et al, which is the largest study about cancer predisposition syndrome in medulloblastoma published to date, there are no findings of MB related to NF1 variants. Actually, only a few case reports are available in literature as detailed in our paper. We have specified this sentence.

Reviewer 2 Report

This is a well worked up case of a medulloblastoma in an NF1 patient. The novel aspect of the case is that there is molecular information and methylation classification as well as good histological description. It is unclear if the medulloblastoma in this case is secondary/driven by NF1 or if this is an accidental tumor in a child with no underlying causation or higher association rate to NF1. 

Author Response

We are pleased that you have considered our work and provided us with very interesting comments that will enhance our paper.

Point 1: This is a well worked up case of a medulloblastoma in an NF1 patient. The novel aspect of the case is that there is molecular information and methylation classification as well as good histological description. It is unclear if the medulloblastoma in this case is secondary/driven by NF1 or if this is an accidental tumor in a child with no underlying causation or higher association rate to NF1.

Response 1: We thank the reviewer for their comments and observation. To date, we do not know if NF1can drive medulloblastoma or if it’s an accidental association. There are only a few cases described in literature that can define this concept. We have added this sentence in the body text: “More cases are needed to determine if NF1 gene mutation or deletions can drive medulloblastoma development”.